# Reprogramming of Amino Acid Metabolism Differs between Community-Acquired Pneumonia and Infection-Associated Exacerbation of Chronic Obstructive Pulmonary Disease

**DOI:** 10.3390/cells11152283

**Published:** 2022-07-24

**Authors:** Haroon Arshad, Anastasios Siokis, Raimo Franke, Aamna Habib, Juan Carlos López Alfonso, Yuliya Poliakova, Eva Lücke, Katina Michaelis, Mark Brönstrup, Michael Meyer-Hermann, Ursula Bilitewski, Jordi Vila, Laurent Abel, Thomas Illig, Jens Schreiber, Frank Pessler

**Affiliations:** 1Research Group Biomarkers for Infectious Diseases, TWINCORE Centre for Experimental and Clinical Infection Research, 30625 Hannover, Germany; dr.haroonarshad@gmail.com; 2Research Group Biomarkers for Infectious Diseases, Helmholtz Centre for Infection Research, 38124 Braunschweig, Germany; 3Department of Systems Immunology and Braunschweig Integrated Centre of Systems Biology, Helmholtz Centre for Infection Research, 38124 Braunschweig, Germany; anastasiossio@gmail.com (A.S.); jc.atlantis@gmail.com (J.C.L.A.); mmh@theoretical-biology.de (M.M.-H.); 4Department of Chemical Biology, Helmholtz Centre for Infection Research, 38124 Braunschweig, Germany; raimo.franke@helmholtz-hzi.de (R.F.); aamna.habib@tuf.edu.pk (A.H.); mark.broenstrup@helmholtz-hzi.de (M.B.); ursula.bilitewski@helmholtz-hzi.de (U.B.); 5Department of Clinical Microbiology, Biomedical Diagnostic Centre (CDB), Hospital Clinic, School of Medicine, University of Barcelona, 08036 Barcelona, Spain; yuliya.poliakova@synlab.es (Y.P.); jvila@clinic.cat (J.V.); 6Clinic for Pneumology, Otto-von-Guericke University, 39106 Magdeburg, Germany; eva.luecke@med.ovgu.de (E.L.); katina_i.m@hotmail.com (K.M.); jens.schreiber@med.ovgu.de (J.S.); 7Laboratory of Human Genetics of Infectious Diseases, Necker Branch, INSERM, 75015 Paris, France; laurent.abel@inserm.fr; 8Imagine Institute, Paris Descartes University, 75015 Paris, France; 9Giles Laboratory of Human Genetics of Infectious Diseases, Rockefeller Branch, Rockefeller University, New York, NY 10065, USA; 10Hannover Unified Biobank, Hannover Medical School, 30625 Hannover, Germany; illig.thomas@mh-hannover.de; 11Centre for Individualised Infection Medicine, 30625 Hannover, Germany

**Keywords:** amino acids, biogenic amines, biomarkers, chronic obstructive pulmonary disease, diagnosis, inflammation, metabolism, pathogenesis, pneumonia, *Staphylococcus aureus*

## Abstract

Amino acids and their metabolites are key regulators of immune responses, and plasma levels may change profoundly during acute disease states. Using targeted metabolomics, we evaluated concentration changes in plasma amino acids and related metabolites in community-acquired pneumonia (CAP, *n* = 29; compared against healthy controls, *n* = 33) from presentation to hospital through convalescence. We further aimed to identify biomarkers for acute CAP vs. the clinically potentially similar infection-triggered COPD exacerbation (*n* = 13). Amino acid metabolism was globally dysregulated in both CAP and COPD. Levels of most amino acids were markedly depressed in acute CAP, and total amino acid concentrations on admission were an accurate biomarker for the differentiation from COPD (AUC = 0.93), as were reduced asparagine and threonine levels (both AUC = 0.92). Reduced tryptophan and histidine levels constituted the most accurate biomarkers for acute CAP vs. controls (AUC = 0.96, 0.94). Only kynurenine, symmetric dimethyl arginine, and phenylalanine levels were increased in acute CAP, and the kynurenine/tryptophan ratio correlated best with clinical recovery and resolution of inflammation. Several amino acids did not reach normal levels by the 6-week follow-up. Glutamate levels were reduced on admission but rose during convalescence to 1.7-fold above levels measured in healthy control. Our data suggest that dysregulated amino acid metabolism in CAP partially persists through clinical recovery and that amino acid metabolism constitutes a source of promising biomarkers for CAP. In particular, total amino acids, asparagine, and threonine may constitute plasma biomarker candidates for the differentiation between CAP and infection-triggered COPD exacerbation and, perhaps, the detection of pneumonia in COPD.

## 1. Introduction

Community-acquired pneumonia (CAP) continues to exert a major burden on populations and health care delivery systems worldwide [1,2]. Despite the wide clinical use of acute phase reactants such as C-reactive protein (CRP) and procalcitonin (PCT) as part of the diagnostic work-up, there is still a great need for more accurate blood biomarkers for CAP, not only to differentiate it from other infectious and non-infectious disorders with similar clinical presentations but also to aid in post-diagnosis treatment decisions such as cessation of antibiotic treatment [3]. The differentiation from chronic obstructive pulmonary disease (COPD) exacerbation presents a clinical challenge in a subpopulation of patients, as COPD flares are often triggered by respiratory infections and may feature symptoms similar or identical to lower respiratory infections. In addition, it is important to identify COPD patients who present with co-existing pneumonia and may require different antibiotic regimes and evaluations, e.g., a heightened awareness of the risk of pneumogenic sepsis. Indeed, COPD is among the most common comorbidities that confer a heightened risk of severe CAP [4].

The importance of amino acids to immune function is most directly explained by their essential roles as building blocks of proteins that affect cellular immune function, which has been studied extensively in the context of malnutrition [5]. However, as reviewed in detail by Yoneda et al. [6] and Zhao et al. [7], amino acid metabolism also plays crucial regulatory roles in a variety of immune responses, including granulocyte function, macrophage activation and polarization, and activation and proliferation of lymphocytes and NK cells. Considering this strong relevance of amino acid metabolism to inflammation and immunity, several studies have addressed changes in amino acid metabolism in CAP (e.g., [8,9,10]) and COPD (e.g., [11,12,13,14,15]). Taken together, these studies have demonstrated marked changes in amino acid metabolism in both disease entities, but a head-to-head comparison of CAP and acute infection-triggered COPD exacerbation, enabling the identification of clinically useful biomarkers to distinguish between these clinically potentially similar entities, has not been performed, and an age- and sex-matched healthy control group is not always included. Moreover, a comprehensive analysis of amino acids and relevant bioactive amino acid metabolites in a single study of CAP or COPD is not available. We have, therefore, measured concentrations of 21 amino acids and 21 amino acid metabolites in plasma samples from patients with CAP, infection-associated COPD exacerbation, and age- and sex-matched control subjects. We find that the two disease entities share certain features such as induction of the Trp-Kyn-NAD+ pathway and downregulation of Val, His, and taurine. Notably, a large drop across all amino acid classes was associated with acute CAP, and several abnormalities upon hospital admission did not normalize at follow-up in spite of clinical recovery. Most importantly, Asn and Thr were regulated in opposite directions in COPD and CAP, which enabled their identification as biomarker candidates for the distinction between the two diagnoses.

## 2. Materials and Methods

### 2.1. Study Design and Study Population

We conducted a prospective, hospital-based study at the University Medical Centre of Otto-von-Guericke-University of Magdeburg, Germany, and the Hospital Clinic of the University of Barcelona, Spain, between March 2014 and November 2015. Details of the study are described in [16], which features a lipidomic analysis of the samples used for the present report. Briefly, we recruited patients with CAP (*n* = 29), COPD exacerbation with infection but without clear evidence of pneumonia (*n* = 13), and age- and sex-matched controls without pulmonary disease or systemic inflammation (*n* = 35). Inclusion and exclusion criteria are summarized in Appendix A. Upon hospital admission, medical history, lung function tests, chest radiographs, vital signs, standard clinical laboratory parameters including an arterial blood gas analysis, and plasma samples for biomarker studies were obtained from all patients. From the CAP patients, clinical data and blood samples were additionally collected on days 2 and 4 and at two follow-up visits after completion of antibiotic treatment (1–14 days, f1; 4–6 weeks, f2). All biosamples were collected according to internationally recognized SOPs and stored at Hannover Unified Biobank according to international biobanking standards. CRP and PCT were measured in all samples in batch after the conclusion of recruitment [16]. The study was approved by the ethics committees of Otto-von-Guericke-University Magdeburg and Hospital Clinic Barcelona.

### 2.2. Mass Spectrometry

We used the AbsoluteIDQ^TM^ p180 kit (Biocrates Life Science AG, Innsbruck, Austria) for targeted metabolomic profiling of plasma, measured on an ultra-high performance liquid chromatography system (Agilent 1290 Infinity) coupled to a triple-quadrupole mass spectrometer (AB Sciex, QTRAP^®^ 6500). The present report comprises an analysis of the 42 amino acids and amino acid metabolites, whereby the amino sulfonic acid taurine was included among the amino acid metabolites. Results of the lipid analysis have been published in [16]. Using the MetIDQ^TM^ RatioExplorer (Biocrates), we also calculated 27 “metabolism indicators” based on these 42 analytes, i.e., sums of analyte subgroups (e.g., glucogenic amino acids) and ratios of analyte subgroups or pairs that allow inferring about enzyme activity, e.g., kynurenine/tryptophan as a measure of indoleamine oxidase 1 (IDO1) activity [17]. Further details are found in [16] and the manufacturer’s application notes (UM_p180_ABSciex_11 and Application Note 1003-1, Biocrates Life Science AG, Innsbruck, Austria; https://biocrates.com (accessed on 15 March 2022)). All samples were processed and frozen within 2 h after blood sampling and thawed only once, i.e., for the present analysis.

### 2.3. S. aureus Infection

Preliminary studies showed that the use of primary cells would not have provided sufficient numbers of cells for the LC-MS/MS measurements. We, therefore, used undifferentiated or phorbol-myristate acetate (PMA)-differentiated myelomonocytic THP1 cells and adenocarcinoma A549 cells. Cells were infected with *S. aureus* strain SH1000 at a multiplicity of infection of 25, incubated at 37 °C for 2 h, repeatedly washed to remove residual medium and analytes adhering to cell membranes, and metabolite concentrations in cells were then measured by LC-MS/MS using the AbsoluteIDQ^TM^ p180 kit (Biocrates). The lipids part of the resulting data set is included in our report on lipid changes in CAP and COPD [16], and further details of the procedure are described therein.

### 2.4. Statistical Analysis

We included only analytes with concentrations > limit of detection (LOD) in ≥75% of all samples. Missing values were subsequently imputed using IBM SPSS version 22, using linear regression models based on age, gender, disease state, and sample collection time point. Data were non-normally distributed, and therefore the Mann–Whitney U test was used to compare differences in median values between two groups, and the Kruskal–Wallis test for differences within CAP across the five time points. Receiver operating characteristic (ROC) curve analysis was used to quantify the discriminatory ability of analytes in paired (binary) comparisons. An area under the ROC curve (AUC) of 1.0 identifies a perfect biomarker and an AUC of 0.5 a nondiscriminatory marker. Robust biomarkers were defined as having an asymptotic significance (*P*) of the ROC curve of <0.05 and a lower bound 95% confidence interval (CI) of >0.5. Linear fitting slope (lfs) modeling was performed as described in [16]. All computational analyses were performed using the respective packages available in Python’s SciPy statistic library (https://docs.scipy.org/doc/scipy/reference/stats.html (accessed on 15 March 2022)), the MNE-Python software package v0.18.1 [18], and the R Foundation for Statistical Computing, version 3.3.2 [19].

## 3. Results

### 3.1. Description of the Study Population

Table 1 summarizes sociodemographic and clinical characteristics of the study groups. There were no differences in age or sex among the three study groups. However, self-reported diabetes was significantly more common in CAP, and self-reported cardiovascular disease and a history of cancer were more common in COPD. A history of lung cancer was not reported. Disease severity in CAP was relatively low as nearly half of CAP patients had a pneumonia severity index (PSI) of I–III. These patients were nonetheless admitted to the hospital at the discretion of the referring or admitting physician. Two of the patients with PSI IV–V developed sepsis, but the clinical outcome was good in all 28/29 (96.5%) CAP patients who presented for follow-up. Fifty percent of COPD patients had GOLD grade IV, indicating a preponderance of higher disease activity as expected among patients with acute exacerbation. Inflammatory markers were markedly higher in CAP than in COPD, and the rate of pathogen detection (41% in CAP and 54% in COPD) roughly corresponded to values expected in clinical practice. Thus, the study population reflected the natural history of acute CAP and COPD exacerbation.

### 3.2. Metabolite Detection

The detected mean concentrations of all analytes in all samples were in the same range as values that were measured in a large cohort of healthy adults using the same mass spectrometric assay [20]. Ala and Gln were the most, and Met and Asp were the least abundant amino acids (Appendix A). Creatinine and taurine were the most, and spermine and serotonin were the least abundant amino acid metabolites, with most having lower concentrations than the amino acids (Appendix A). Out of the 21 amino acids and 21 amino acid metabolites, 20 amino acids (95%) and 10 amino acid metabolites (48%) were detected >LOD in ≥75% of the samples and were thus used for the subsequent analyses (Appendix A).

### 3.3. Reprogramming of Amino Acid Metabolism Differs between CAP and COPD and Tends to Normalize with Clinical Improvement of CAP

A principal component analysis (PCA) showed that changes in amino acid populations were greatest in acute CAP (d1) and appeared to normalize across the time course, although there was still some difference between CAP (f2) and Controls. Changes in COPD were substantially less pronounced (Figure 1a). Amino acid metabolite changes followed a similar pattern in CAP, but changes in COPD were more substantial as they differed more strongly from Controls and CAP follow-ups (Figure 1b). The metabolism indicators mostly reflected the normalization in CAP at follow-up and smaller changes in COPD (Figure 1c). A Euclidian distance analysis confirmed that amino acid reprogramming was markedly greater in CAP than in COPD but that reprogramming of amino acid metabolites was somewhat greater in COPD (Figure 1d).

### 3.4. Decreased Amino Acid Concentrations Are the Hallmark of Acute CAP

Differential abundance analysis revealed a strong tendency in acute CAP towards decreased concentrations of most amino acids (Figure 2a), whereas only one amino acid (Phe) was upregulated. The Trp catabolite kynurenine (Kyn) was the most significantly upregulated amino acid metabolite. Of note, whereas most parameters tended to normalize with clinical recovery, it was evident that some changes persisted through f2. Changes in COPD were strikingly different in that there was a balance between up- and downregulated amino acids; in fact, some amino acids that were downregulated in CAP were upregulated in COPD (Asn, Thr, and Met). However, the higher Met levels in COPD could also be explained by the higher frequency of patients with cardiovascular disease in this group (Appendix A). Changes in amino acid metabolites in COPD vs. Controls differed from CAP, particularly by higher ADMA levels, but this could also be explained by the higher frequency of patients with cardiovascular disease in COPD (Appendix A).

### 3.5. Persistence of Abnormalities in Amino Acid Metabolism during Resolution of CAP

Figure 2a shows that six parameters (ADMA, Glu, Orn, His, Trp, and Cit) did not normalize with resolution of CAP (f2). Four of these (Cit, Trp, His, and Orn) were depressed on day 1 and did not reach normal values by f2. SDMA levels remained elevated from d1 through f2, but this could be explained by the higher frequency of patients with diabetes in this group (Appendix A). Glutamate presented a notable phenomenon in that its levels were slightly depressed upon hospital admission, normalized quickly, and then continued to rise above control levels through convalescence. Thus, key aspects of amino acid metabolism did not normalize even when the patients had reached a state of clinical convalescence.

Changes in the metabolism indicators confirmed the broad downregulation of amino acids in CAP and, in addition, revealed Kyn/Trp ratio as the most highly increased value, suggesting induction of the Trp-Kyn-NAD+ pathway (Figure 2c). However, it also revealed a shift in arginine metabolism in CAP in that there was less catabolism towards citrulline (decreased Cit/Arg ratio) but a preference toward conversion to ADMA and SDMA, as evidenced by increased ratios of ADMA, SDMA, and total DMA in relation to Arg. This was only partially explained by higher SDMA levels in CAP patients with diabetes, as ADMA levels did not differ between CAP patients with or without diabetes. In contrast, in COPD, the ADMA and SDMA ratios tended to be lower than in Controls (Figure 2c,d). Thus, this analysis substantiated the observations from the PCA that changes in amino acid metabolism differed greatly between CAP and COPD and also suggested that some functional networks are regulated in opposite directions in the two diagnoses.

### 3.6. Comparisons with Amino Acid Metabolic Changes in S. aureus-Infected Human Cell Lines

In order to test whether the pronounced changes in CAP might be at least partially due to changes in host cells infected with a pathogen or exposed to mediators released from infected cells, we applied the same metabolomic analysis to A549 cells (representing airway epithelial cells), THP1 cells (resembling circulating monocytes), and differentiated THP1 cells (resembling tissue macrophages) that were infected with *S. aureus*, a common CAP pathogen. Among the three cell types, regulation patterns were most similar between THP1 and dTHP1 cells in that there was a strong tendency toward down-regulation of both analyte classes. In contrast, in A549 cells, 20 analytes were upregulated, but only four and one in THP1 and dTHP1 cells, respectively (Figure 3a,b). There was considerable agreement between CAP and THP1 and dTHP1 cells in terms of downregulated analytes; however, there was no apparent agreement between the five analytes that were upregulated in CAP and those that were upregulated in the cell lines (Figure 3c,d). These results suggest that changes in infected host cells may contribute to the observed reduction in plasma amino acids and their metabolites but that the bulk of changes originates from other processes.

### 3.7. Identification of Diagnostic Biomarker Candidates

ROC analysis was then used to identify diagnostic biomarker candidates in plasma. For the distinction CAP vs. Controls, eight analytes exceeded the cut-off of AUC = 0.8 for “excellent classification” [21]. The three most accurate biomarkers were Trp, His, and Ala, with AUC of 0.96, 0.94, and 0.93, respectively (Figure 4a). Biomarker potential for COPD vs. Controls was substantially lower in that only two analytes with AUC >0.8 were identified (ADMA, 0.81; taurine, 0.80), but both had lower CI of AUC crossing below 0.5 (Figure 4b), and the higher ADMA levels in COPD were likely due to the higher frequency of cardiovascular disease in this group (Appendix A). Surprisingly, there were 12 biomarkers with AUC >0.8 for the distinction between CAP and COPD, the best ones being Asn (AUC 0.92), Thr (AUC 0.92), and Met (0.89) (Figure 4c). All three were regulated in opposite directions in CAP and COPD with respect to Controls, i.e., decreased in CAP but increased in COPD (Figure 2a). Of these, the increased Met concentrations in COPD were also explained by the higher frequency of cardiovascular disease in this group (Appendix A). Considering the small size of the COPD group, we assessed the robustness of the identified biomarkers for CAP vs. COPD with an oversampling procedure where *n* was increased to that of CAP, i.e., 29. Essentially the same biomarkers were identified, but robustness increased in that *p* values decreased further and CIs no longer crossed below 0.5 (Appendix A). A similar pattern of differences in biomarker potential was observed in the metabolism indicators (Figure 4d–f): the most accurate biomarkers were found for CAP vs. Controls (*n* = 5) and CAP vs. COPD (*n* = 5), but only one (Thr/Ser ratio) for COPD vs. Controls. Of note, elevated Kyn/Trp ratio, indicating increased activity of IDO1 (the key enzyme of the extrahepatic Trp-Kyn-NAD+ pathway), was the most accurate (AUC 0.94) biomarker for CAP vs. Controls. Total amino acids were the best metabolism indicator biomarker for CAP vs. COPD (AUC 0.93), reflecting the impact of the marked decrease in concentrations of the majority of amino acids in acute CAP. Figure 4g–l shows the measured values in the three diagnostic groups (CAP on d1 only) of the best metabolite and metabolism indicator biomarker for the three paired comparisons.

### 3.8. Changes in Amino Acid Metabolism That Correlate with Resolution of CAP

The differential abundance analysis (Figure 2) had shown that some changes in CAP persisted through clinical recovery but that, overall, there was a tendency toward normalization. Considering the need for biomarkers that correlate with resolution of CAP, we applied a linear fitting slope (LFS) vs. 1-normalized distance to fitting (NDF) analysis to identify analytes and metabolism indicators that were most closely associated with clinical recovery (normalization at f1 and f2) and resolution of systemic inflammation (CRP). This analysis identified Pro and 4OH-proline (the result of in-protein hydroxylation of Pro) as the best correlating analytes and Kyn/Trp ratio and glucogenic amino acids as the best correlating metabolism indicators (Figure 5).

### 3.9. Correlations with Systemic Inflammation

Spearman’s correlation analysis was performed between the analytes and CRP and PCT in all CAP samples. Consistent with the predominance of amino acid downregulation in CAP, most correlations of amino acids were negative, with Trp correlating the most strongly (r = −0.59) (Figure 6a). Phe was the notable exception, as it correlated positively with both CRP (r = 0.58) and PCT (r = 0.3). Only negative significant (*p* ≤ 0.05) correlations were observed between the amino acid metabolites and CRP, whereby 4OH-Pro correlated the most strongly (r = −0.41), whereas both positive and negative correlations, albeit weaker than with CRP, were observed with PCT. Both positive and negative correlations were observed for metabolism indicators, with Asn/Gln ratio being the most positive (r = 0.47) and Tyr/Phe the most negative (r = −0.59). Of note, 19 out of 62 (31%) correlations of single metabolites with either CRP or PCT were not significant at *p* ≤ 0.05, and only three (5%) had correlation coefficients >0.5. Taken together, these results suggest that the observed changes in amino acid metabolism in CAP are only partially due to systemic inflammation.

## 4. Discussion

We compared changes in plasma amino acids and amino acid metabolites between CAP and infection-triggered COPD exacerbation, thus presenting the first head-to-head comparison of amino acid metabolism in these clinically potentially similar but pathophysiologically fundamentally different disorders.

Regarding changes uniquely or preferentially seen in CAP, we found pronounced, nearly universal decreases in amino acid concentrations in acute CAP. The question arises whether this may be due to decreased food intake in an acute illness such as CAP. However, we consider this unlikely because the same effect would be expected in infection-triggered COPD exacerbation, as the overall well-being of the patients is at least as compromised as in CAP. It has been known for some time that systemic infections reprogram protein metabolism toward a catabolic state [22]. Moreover, a similar broad decrease in amino acid concentrations was seen in *S. aureus*-infected THP1 cells (Figure 3). We, therefore, favor a model whereby the nearly universal drop in plasma amino acids in CAP results from the interplay between a systemic shift towards catabolism and changes due to reprogramming of metabolic pathways involving amino acids in infected cells or in uninfected cells that are exposed to the altered cytokine milieu.

Phe was the only amino acid whose concentration increased in CAP, and Tyr/Phe ratio was among the most decreased metabolism indicators in CAP. This agrees with earlier reports of increased Phe accumulation in inflammation (reviewed in [23]), which is postulated to be due to decreased activity of phenylalanine 4-hydroxylase as the result of limiting concentrations of the cofactor 5,6,7,8-tetrahydrobiopterin (BH4). Since Phe is also elevated in mood disorders with depressive features, these authors suggested that increased Phe contributes to the overall feeling of mental unwellness in patients with acute or chronic inflammation.

Activation of the Trp-Kyn-NAD+ pathway was an important shared feature of CAP and COPD exacerbation, and the fact that activation was greater in CAP agreed well with the higher degree of systemic inflammation found in this disorder. Rapoport et al. showed in 1970 that Trp catabolism is enhanced in infections in humans [24], and induction of the Trp-Kyn-NAD+ pathway has since then been linked to immune activation and modulation in a variety of infectious and non-infectious inflammatory disorders (e.g., [25,26]). The close association between normalization of the Kyn/Trp ratio and resolution of CAP in our study underscores the previously identified importance of activation of this pathway to disease severity and long-term prognosis in CAP [10,27]. Zinellu et al. described elevated Kyn levels and activation of the Trp-Kyn-NAD+ pathway as a feature of mild-to-moderate COPD [13], but ours is the first study to show that, overall, the degree of activation is greater in CAP. A similar situation was seen with respect to Arg catabolism in that reprogramming of the respective pathway was evident in both CAP and COPD but was clearly more dramatic in CAP. Arg can be catabolized to citrulline, releasing NO, which has vaso- and bronchodilatory properties but can also be turned into cytotoxic reactive nitrogen compounds ([9,15] and references therein). A shift towards the formation of ornithine and (post-translationally) SDMA or ADMA can reduce the available pool of Arg and the subsequent formation of NO. We observed a preferential increase in ADMA in COPD and SDMA in CAP, and Cit/Arg and Cit/Orn were similar in the two groups. However, Arg concentrations were uniquely reduced in CAP, and all three dimethylarginine ratios with respect to Arg were substantially higher in CAP than in COPD, suggesting much more pronounced sequestration of free Arg in CAP. Vögeli et al. detected an association between elevated SDMA and ADMA concentrations and disease severity and 6-year mortality in CAP but argued that this was mostly a confounding effect of age and comorbidities [15]. Our data agree with this notion insofar as none of the dimethyl-arginine associated parameters correlated appreciably with resolution of CAP (Figure 2). Indeed, the significant increase in SDMA in acute CAP was at least partially explained by its association with diabetes in this group. Raised ADMA levels are associated with cardiovascular disease, and the frequency of heart disease was significantly higher in our COPD group than in Controls and CAP (Table 1). This comorbidity may thus have contributed to the raised ADMA levels in COPD. However, several studies have demonstrated increased blood ADMA concentrations in COPD, especially in exacerbation, and revealed it as a marker of increased airway resistance [12,15,28,29,30]. Further studies are needed to clarify the role of ADMA in COPD pathogenesis. In particular, it remains to be explained whether increased ADMA levels in comorbidities such as cardiovascular disease have a direct impact on the severity of COPD.

Concentrations of taurine were significantly decreased in both CAP and COPD. Taurine is a product of cysteine catabolism and is the most abundant free amino acid in humans [31]. It possesses a variety of properties such as reduction of oxidative stress and acute inflammation [31,32]. It is thus tempting to speculate that a reduction of taurine contributes to clinical symptoms and, perhaps, organ dysfunction in CAP and COPD patients with sufficiently reduced levels.

The decreased His levels seen in both CAP and COPD have also been reported in other inflammatory disorders, e.g., inflammatory bowel disease [33]. They may stem in part from increased activity of histidine decarboxylase, which catabolizes His to its biogenic amine histamine not only in mast cells but also in other immune and non-immune cells. Activation of this enzyme has been shown in murine models of sepsis and pneumonia and in urate crystal inflammation, which shares central features with inflammation seen in invasive bacterial infections [34,35,36,37].

Increased concentrations of Asn, Lys, and Thr were salient features of infection-triggered COPD exacerbation. Since significantly more patients with COPD than with CAP were treated with corticosteroids, we posed the question whether this might be a corticosteroid effect. However, the impact of corticosteroid treatment on blood amino acid levels appears to be different, and the increases in Ala or Gln that have been described in the literature [33,38] were not seen in our COPD group. Their reciprocal regulation in CAP vs. COPD identified Asn and Thr as highly accurate diagnostic biomarkers for the differentiation between the two entities. Elevation of these amino acids has not been described in COPD exacerbation (or cardiovascular disease, as the major comorbidity in our study), and the pathophysiological significance is uncertain. It should be noted, though, that Inouye et al. [14] performed a study comparing plasma amino acid levels between patients with acute COPD exacerbation with vs. without bacterial infection. According to their inclusion criteria, the bacterial infection group in that study is much more likely to have a lower respiratory infection than our COPD group. Intriguingly, their bacterial infection group displayed all amino acid changes seen in our CAP group but not increased Asn or Thr. Considering that a decrease in total amino acid was the best metabolism indicator biomarker for CAP vs. COPD, our results suggest that plasma levels of total amino acids and of Asn and Thr merit further evaluation as potential biomarkers to distinguish between CAP and infection-associated COPD exacerbation and to stratify patients with COPD exacerbation according to the likelihood of having co-existing pneumonia.

Our study is limited by the small size of the COPD group, making it difficult to draw more general conclusions. This is complicated further by the fact that there are no published data from COPD patients with identical inclusion criteria. Nonetheless, the oversampling analysis, in which we modeled increasing the sample size of the COPD group, provided some additional validity to the COPD data. Clearly, further studies should be conducted to validate the clinically most relevant findings.

In summary, this first comparison of amino acid metabolism in acute CAP vs. infection-associated COPD exacerbation revealed profound common and unique changes in these disease entities, identified promising biomarker candidates for the differentiation between the two, and suggests that a global reduction of plasma amino acid concentrations is a cardinal feature of acute CAP.

## Figures and Tables

**Figure 1 cells-11-02283-f001:**
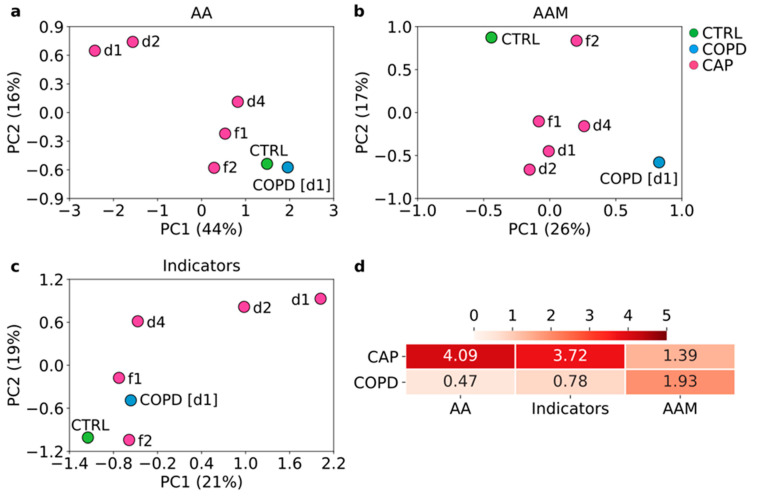
**Reprogramming in acid metabolism in acute CAP normalizes with clinical recovery.** Principal component analysis (PCA) based either on concentrations of the 20 amino acids or 10 amino acid metabolites that were detected >LOD in at least 75% of all samples (**a**,**b**) or on the 29 metabolism indicators (**c**). (**d**) Mean Euclidian distances on day 1 between CAP or COPD vs. Controls, based either on amino acids, amino acid metabolites, or metabolism indicators. Reprogramming is greatest in CAP for amino acids and metabolism indicators but is slightly greater in COPD for amino acid metabolites. AA—amino acids; AAM—amino acid metabolites.

**Figure 2 cells-11-02283-f002:**
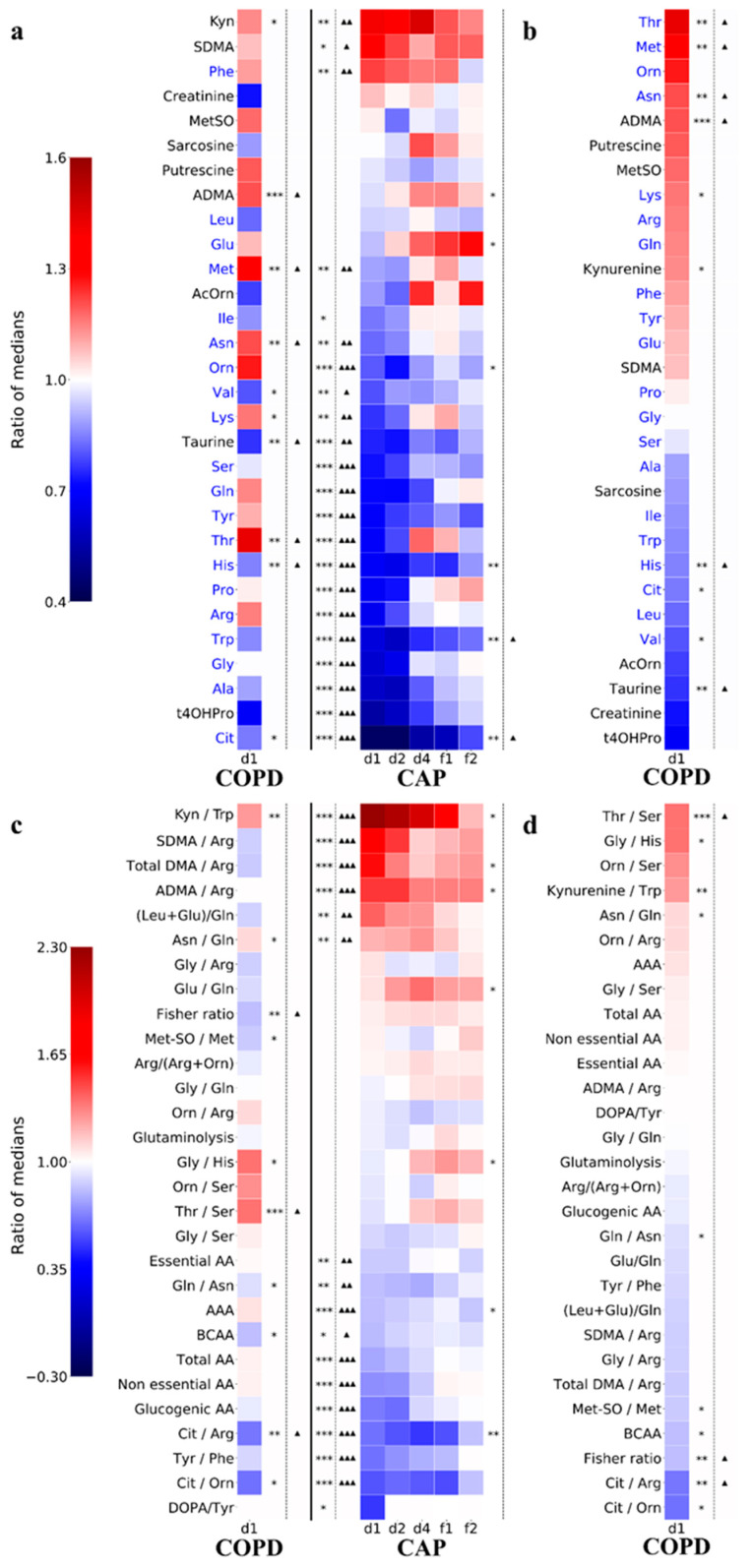
**Marked decrease in plasma amino acid concentrations in CAP.** Analysis based on the same 20 amino acids and 10 amino acid metabolites, and 29 metabolism indicators as in Figure 1. (**a**,**b**) Analysis based on amino acids and amino acid metabolites. The analytes are arranged in descending order according to ratio of median concentrations CAP d1/Controls (“fold change”). (**b**) Same analysis as A, but showing differential abundance in COPD only, with analytes arranged in descending order according to fold change COPD/Controls. (**c**) Analysis based on the 29 metabolism indicators. The indicators are ranked vertically according to descending ratio of median values in CAP d1/Controls. (**d**) Same analysis as C, but showing differential abundance in COPD only, arranged in descending order according to fold change COPD/Controls. Differential abundance is expressed on a linear scale by the color scheme in the legends shown in A and C. Analyte classes are identified by font color: black = amino acids, blue = amino acid metabolites. * *p* < 0.05, ** *p* < 0.01, *** *p* < 0.001 (Mann–Whitney U test). Δ *p* < 0.05, ΔΔ *p* < 0.01, and ΔΔΔ *p* < 0.001 (FDR-corrected). AA—amino acids; AAM—amino acid metabolites. Abbreviations of the metabolism indicators are defined in Appendix A.

**Figure 3 cells-11-02283-f003:**
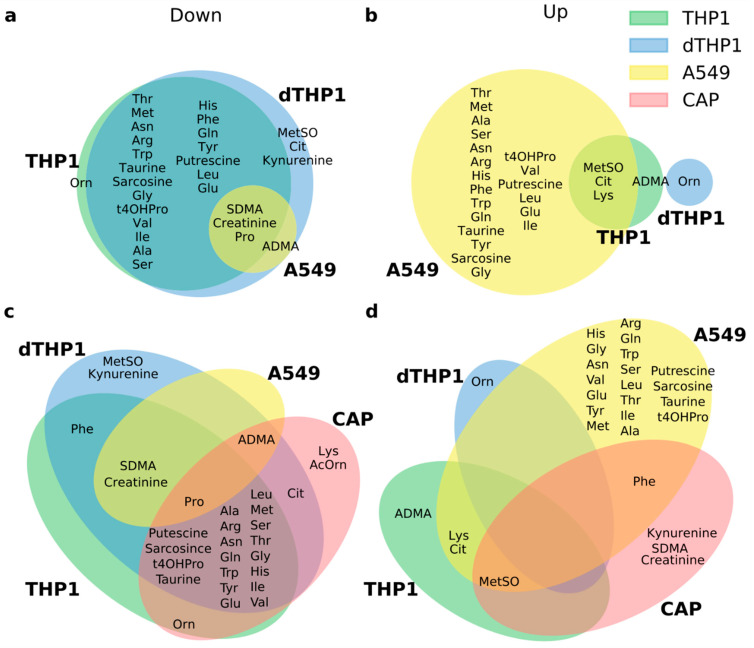
**Common and distinct features of changes in amino acid metabolism: comparison of acute CAP with three human cell lines infected with *S. aureus*.** A549, THP1, and PMA-differentiated THP1 (dTHP1) cells were infected with *S. aureus* strain SH1000 at an MOI of 25, harvested 2 h post infection, and analyte concentrations in cells were measured with the same targeted metabolomic assay as used for the plasma samples. Shared and common differentially abundant (FDR ≤ 0.05) analyte subpopulations are defined by the intersects of the circles/ovals in the Venn diagrams. (**a**,**b**) *S. aureus*-infected cell lines. (**c**,**d**) acute CAP and *S. aureus*-infected cell lines. The analyte abbreviations are defined in Appendix A.

**Figure 4 cells-11-02283-f004:**
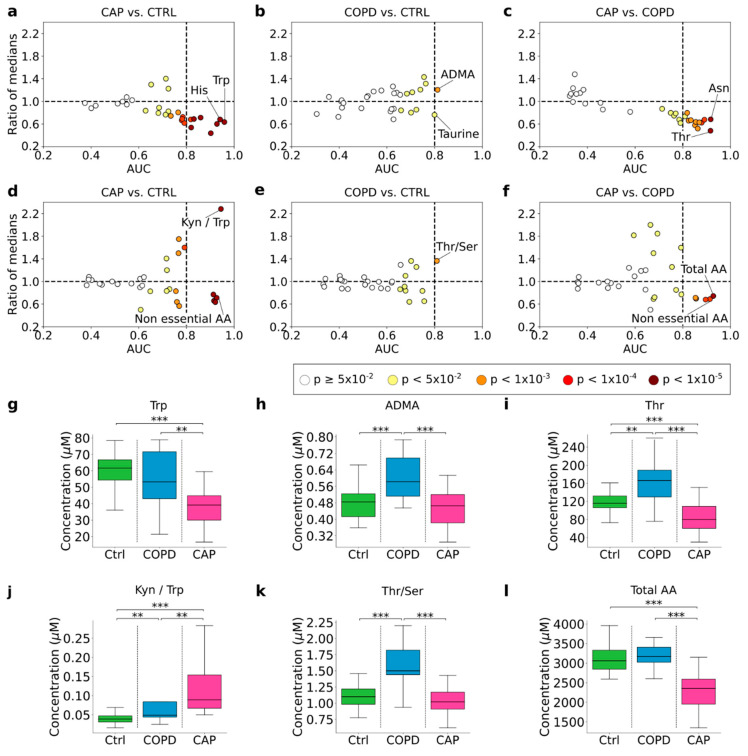
**Quantitative** (**ROC curve**) **biomarker analysis of plasma amino acids and amino acid metabolites in acute CAP and COPD.** Differential abundance (ratio of medians, “fold change”; *y*−axis) is plotted against area under the ROC curve (AUC, *x*−axis), with fill color darkness indicating the asymptotic significance of the ROC curve. Each dot corresponds to one analyte (**a**–**c**) or one metabolism indicator (**d**–**f**). The dotted vertical line marks AUC 0.80, the cut−off for “excellent classification” [21]. (**a**,**d**) CAP baseline vs. Controls; (**b**,**e**) COPD vs. Controls; (**c**,**f**) CAP baseline vs. COPD. The ROC analysis parameters of all markers with AUC ≥0.8 for each comparison are listed in Appendix A. (**g**–**i**) Concentrations of the best amino acid or amino acid metabolite biomarkers for each of the comparisons shown in A-C: CAP vs. Ctrl = tryptophan (Trp); COPD vs. Ctrl = asymmetric dimethyl arginine (ADMA); CAP vs. COPD = threonine (Thr). (**j**–**l**) Values of the best metabolism indicator biomarkers for each of the comparisons shown in (**d**–**f**): CAP vs. Ctrl = kynurenine/tryptophan ratio; COPD vs. Ctrl = threonine/serine ratio; CAP vs. COPD = total amino acids. Box = 25th–75th percentile; lower whiskers = bottom quartile; upper whiskers = upper quartile; horizontal line = median. ** *p* < 0.01, *** *p* < 0.001 (Mann–Whitney U test).

**Figure 5 cells-11-02283-f005:**
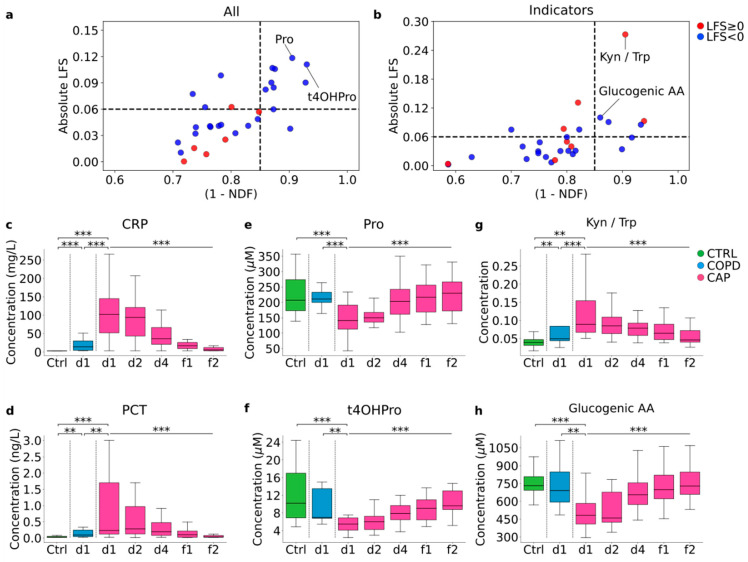
**Selection of biomarkers that correlate with resolution of CAP.** Amino acids and amino acid metabolites (**a**) and metabolism indicators (**b**) that correlate best with resolution of CAP were identified using a plot of linear fitting slope (*y*−axis, LFS) vs. 1−normalized distance to fitting (*x*-axis, NDF). (**c**–**h**) Dynamics of change throughout the time course: CRP (**c**)**,** PCT (**d**)**,** the two best correlating amino acids/amino acid metabolites (proline (**e**), 4OH-proline (**f**)), and the two best correlating metabolism indicators, Kyn/Trp ratio (**g**) and glucogenic amino acids (**h**). ** *p* < 0.01, and *** *p* < 0.001 (Mann–Whitney U test for paired comparisons, Kruskal–Wallis H-test for CAP d1 through CAP f2). AA—amino acids.

**Figure 6 cells-11-02283-f006:**
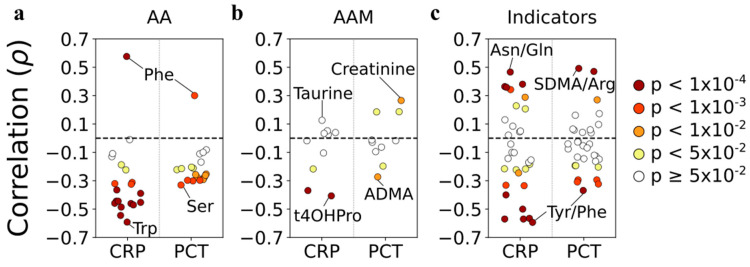
**Correlations between amino acids, amino acid metabolites, and metabolism indicators and CRP and PCT in CAP.** The analysis is based on values measured in all CAP samples (d1-f2). *Y*−axis values correspond to Pearson’s rho; *p* values are indicated by the fill color of the circles. (**a**) Amino acids. (**b**) Amino acid metabolites. (**c**) Metabolism indicators. AA—amino acids. AAM—amino acid metabolites.

**Table 1 cells-11-02283-t001:** Sociodemographic and clinical characteristics.

	CAP (*n* = 29)	Infection-Associated COPD Exacerbation (*n* = 13)	Controls (*n* = 33)	*p* Value	
				All Groups ^a^	CAP vs. COPD ^b^
**Demographics**					
Female (%)	38	46	36	0.30	0.25
Male (%)	62	54	64		
Median age (range)	60 (24–90)	62 (55–81)	59 (24–90)	0.78	0.31
**Medical history** (past or current)					
Diabetes (%)	38	17	24	2.0 × 10^−3^	9.0 × 10^−4^
Cardiovascular disease (%)	28	50	24	1.6 × 10^−4^	1.4 × 10^−3^
Cancer (%)	10	30	6	< 1.0 × 10^−5^	4.1 × 10^−4^
**Treatment** (at time of first blood sample)					
Antimicrobial (%)	100	100	0	NA	NA
Corticosteroid (%)	22	62	3.3	< 1.0 × 10^−5^	< 1.0 × 10^−5^
**Disease severity**					
	PSI risk class (%)	GOLD grade (%)			
	I = 21	I = 0	NA	NA	NA
	II = 20	II = 25	NA	NA	NA
	III = 14	III = 25	NA	NA	NA
	IV = 24	IV = 50	NA	NA	NA
	V = 21	NA	NA	NA	NA
**Laboratory results**					
CRP (mg/L, ref. < 5 mg/L)	102 (3.1–428)	14 (3.1–91)	3.1 (3.1–7.1)	2.5 × 10^−12^	2.5 × 10^−5^
PCT (ng/l, ref. < 0.5 ng/L)	0.23 (0.02–38)	0.09 (0.02–1.0)	0.02 (0.02–0.08)	5.0 × 10^−9^	7.1 × 10^−3^
Pathogen detected (%)	41	54	NA	NA	NA

Adapted from Arshad H, *J Transl Med* 2019, 17, 365. Copyright by the authors. ^a^ Kruskal–Wallis test ^b^ Mann–Whitney U test (continuous variables) and Chi^2^ test (categorical variables).

## Data Availability

The data are available from the corresponding author upon a reasonable request.

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
