# Peer review of "Reprogramming of Amino Acid Metabolism Differs between Community-Acquired Pneumonia and Infection-Associated Exacerbation of Chronic Obstructive Pulmonary Disease"

_cells, 2022, doi:10.3390/cells11152283_

Round 1

Reviewer 1 Report

Congratulations for your hard work!

The article regarding reprogramming of amino acid metabolism differs between community-acquired pneumonia and infection-associated exacerbation of chronic obstructive pulmonary disease is an interesting one not only for clinical purposes, but also for clinical research. The topic is original and adds value to the subject area.

The article represents a future point of view in medicine and identifying biomarkers that can be used routinely later is auspicious in medical practice.

The paper is very well written, with a strong and clear data and statistical analysis.

The conclusions are consistent with the evidence and arguments presented.

Author Response

We thank the reviewer for the positive evaluation!

Reviewer 2 Report

Arshad et al. performed a relatively intact study by analyzing the plasma amino acid concentration changes and related metabolites in community-acquired pneumonia (CAP) from the patients under convalescence, aiming to identify several amino acids that could be utilized as biomarker candidates in the clinic to distinguish CAP and COPD. The manuscript demonstrated that the current research has potential clinical value. Some concerns should be addressed.

1. For table 1, the cancer part. Suppose the cancer group was divided into respiratory system cancer group and non-respiratory system cancer. Will it make a difference compared to the current cancer group results?

2. Why did the authors choose cancer cell lines (A549, THP1 and dTHP1) instead of normal cell lines like HSAEC to perform the studies shown in figure 3? Based on table1 data, cancer is one of the significant clinical characteristics. Will there be any difference/commence between cancer cell lines and normal/immortalized cell lines in amino acid metabolic changes in S. aureus infection?

Author Response

The reason for for using cell lines was that the LC-MS/MS measurements require a lot of cells (several mio. per sample). It would therefore not have been feasible to establish the assays and perform the assays with primary human cells.  We added a sentence about this to Methods. (lines 132-133)

There were no cases of lung/respiratory tract cancer. We added this to Results. (lines 163-164).  The infections were meant to mimick responses in CAP, not a difference between CAP vs. COPD, and CAP is the disease with a low frequency of cancer (10 vs. 30% in COPD). We therefore do not believe that we are reinforcing a "cancer bias" in  the study by using these cell lines. 

Unrelated, we also adapted the discussion about ADMA in COPD to give more credit to the notion that it may be a mediator between a comorbidity and disease severity. (lines 421-424)